# Impact of labor characteristics on maternal and neonatal outcomes of labor: A machine-learning model

Sherif A. Shazly[1], Bijan J. Borah[1,2,3], Che G. Ngufor[3,4], Vanessa E. Torbenson[1], Regan N. Theiler[1], Abimbola O. Famuyide [1] *

1 Department of Obstetrics and Gynecology, Mayo Clinic, Rochester, Minnesota, 2 Division of Health Care Delivery Research, Mayo Clinic, Rochester, Minnesota, 3 The Robert D. and Patricia E. Kern Center for the Science of Health Care Delivery, Mayo Clinic, Rochester, Minnesota, 4 Department of Artificial Intelligence and Informatics, Mayo Clinic, Rochester, Minnesota

* famuyide.abimbola@mayo.edu

## Abstract

### Introduction

Since Friedman's seminal publication on laboring women, numerous publications have sought to define normal labor progress. However, there is paucity of data on contemporary labor cervicometry incorporating both maternal and neonatal outcomes. The objective of this study is to establish intrapartum prediction models of unfavorable labor outcomes using machine-learning algorithms.

### Materials and methods

Consortium on Safe Labor is a large database consisting of pregnancy and labor characteristics from 12 medical centers in the United States. Outcomes, including maternal and neonatal outcomes, were retrospectively collected. We defined primary outcome as the composite of following unfavorable outcomes: cesarean delivery in active labor, postpartum hemorrhage, intra-amniotic infection, shoulder dystocia, neonatal morbidity, and mortality. Clinical and obstetric parameters at admission and during labor progression were used to build machine-learning risk-prediction models based on the gradient boosting algorithm.

### Results

Of 228,438 delivery episodes, 66,586 were eligible for this study. Mean maternal age was 26.95 ± 6.48 years, mean parity was 0.92 ± 1.23, and mean gestational age was 39.35 ± 1.13 weeks. Unfavorable labor outcome was reported in 14,439 (21.68%) deliveries. Starting at a cervical dilation of 4 cm, the area under receiver operating characteristics curve (AUC) of prediction models increased from 0.75 (95% confidence interval, 0.75–0.75) to 0.89 (95% confidence interval, 0.89–0.90) at a dilation of 10 cm. Baseline labor risk score was above 35% in patients with unfavorable outcomes compared to women with favorable outcomes, whose score was below 25%.

**Data Availability Statement:** Our data set belongs to the NICHD. We have attached a copy the agreement we had with NICHD. To access these

data sets, please reach out directly to the NICHD: NICHD DASH Administrator https://dash.nichd.nih. gov It is titled " Consortium of Safe Labor" datasets from the NICHD. I can confirm authors did not have special privileges not available to others who apply for access to the data.

**Funding:** The author(s) received no specific funding for this work.

**Competing interests:** The authors have declared that no competing interests exist.

## Conclusion

Labor risk score is a machine-learning–based score that provides individualized and dynamic alternatives to conventional labor charts. It predicts composite of adverse birth, maternal, and neonatal outcomes as labor progresses. Therefore, it can be deployed in clinical practice to monitor labor progress in real time and support clinical decisions.

## Introduction

Management recommendations of labor and delivery evolve constantly to accommodate evidence from literature. A major conundrum every obstetrician faces in managing women in labor is weighing maternal and neonatal risks of delayed intervention against risks of unindicated caesarean delivery (CD). Although the incidence of CD has substantially increased in the past 3 decades [1], there has been no discernible decline in maternal or neonatal adverse outcomes [2]. Labor dystocia represents the most common indication for primary CD, but diagnosis of labor dystocia lacks consistent evidence-based and globally acceptable definition. This may contribute, in part, to the increases in rates of CDs [3].

One of the earliest studies to define normal labor progress was conducted in 1955 by Friedman [4]; based on observation of 500 women in labor, Friedman described the normal course of labor, known as the "Friedman curve". The World Health Organization (WHO) relied on Freidman's data in its construction of a labor partogram for managing labor and labor dystocia, particularly in low-resource countries. A recent Cochrane review failed to demonstrate a significant difference in the rate of CD among women who were or were not managed by the WHO partogram [5]. In 2002, Zhang et al [6] studied 1,329 term nulliparous parturient and suggested that the Friedman curve may not be reflective of contemporary labor progress patterns. To verify their hypothesis, Zhang and colleagues conducted a multicenter study that prospectively collected clinical data, including pre-labor characteristics, intrapartum parameters, and maternal and neonatal outcomes of women who delivered at 1 of 12 studied clinical centers across the United States. This has become known as the *Consortium on Safe Labor*" database [7]. This study created a new labor curve, which substantially modified the management of labor in the United States. However, the authors' analysis specifically excluded maternal and fetal outcomes data; thus, questions have been raised about the impact of adopting the Zhang labor curve on CD rates as well as maternal and fetal outcomes [8]. The objective of this study is to establish an individualized labor chart, through a series of intrapartum prediction models, using machine-learning algorithms that incorporate data on CD and obstetric outcomes. This dynamic tool may facilitate patient counselling and decision making and reduce the rate of CD, maternal, and neonatal complications.

## Materials and methods

The protocol of the current study was reviewed and approved by DASH prior to acquisition of CSL anonymized database. The current study did not require patient contact or new data collection. Thus, institutional review board approval/patient consents were not applicable to the current study. Original data was collected retrospectively by 19 hospitals. All contributing hospitals to this database obtained institutional review boards according to the primary study (Zhang et al 2010).

## Study population

The Eunice Kennedy Shriver National Institute of Child Health and Human Development (NICHD) established the NICHD Data and Specimen Hub (DASH), which is a database sharing provider data that enables investigators to use de-identified data from NICHD-funded research studies for the purpose of research. A consortium of 12 clinical centers located in all 9 districts of the American College of Obstetricians and Gynecologists provided electronic obstetric, labor, and newborn data between 2002 and 2008, which created a large database, known as the Consortium on Safe Labor database. This database was used by Zhang et al [7] to create the contemporary labor curves published in 2010. This database includes 228,438 deliveries with a total of 779 antepartum, intrapartum, and postpartum variables. The de-identified version of this database was obtained with permission through a DASH data use agreement for the purpose of this study.

## Study outcomes

The aim of the current study is to develop a series of intrapartum models that comprise baseline variables and dynamic (intrapartum) variables to predict the probability of unfavorable labor outcome (labor risk score [LRS]). *Unfavorable labor outcome* is defined as the composite of 1 of the following unfavorable outcomes: unsuccessful vaginal delivery (CD in active labor), postpartum hemorrhage (defined as estimated blood loss >1,000 mL) or need for transfusion of blood products, suspected or confirmed intra-amniotic infection (IAI), shoulder dystocia, admission to the neonatal intensive care unit (NICU), Apgar score below 7 at 5 minutes, umbilical arterial pH below 7.00, neonatal hypoxemic ischemic encephalopathy, neonatal ventilation use or continuous positive airway pressure therapy, neonatal intracranial hemorrhage, neonatal sepsis, or neonatal death. LRS is a term that describes the probability of unfavorable labor outcome, as calculated by the model.

To accommodate the objectives of this study, women with multifetal pregnancy, intrauterine fetal death, preterm labor (defined as birth at <37 weeks of gestation), fetal anomalies, or women who underwent elective CD, CD for failed induction, fetal malpresentation, cord prolapse, active herpetic lesion, CD performed prior to the onset of active labor (CD at cervical dilation of ≤5 cm), and women with 3 or more prior CDs (history of CD) were excluded. Women with inadequate documentation, defined as documentation of less than 2 cervical examinations, were also excluded from the study.

## Prediction models

A set of prediction models were established to predict the primary outcome of this study. A baseline model was created using variables identified at the time of patient admission (baseline predictors). A series of intrapartum prediction models were set up to incorporate dynamic variables determined by pelvic examination starting at a cervical dilation of 4 cm and other parameters, including use of oxytocin to augment labor and meconium-stained amniotic fluid. These variables included current cervical dilation, cervical effacement (categorized as 0%-30%, 40–50%, 60–70%, or 80% or more), head station (categorized as –3, –2, –1, 0, +1, or +2), time interval between current and previous examinations, change in cervical dilation between current and previous examinations, and dilation delta (defined as change in cervical dilation from previous examination divided by time interval between the 2 examinations). Intrapartum variables that could not be linked to a particular cervical dilation (e.g., meconium-stained amniotic fluid) were incorporated into the 10-cm prediction model. Although intrapartum fetal heart rate monitoring was considered in study protocol, it was not included in these models due to lack of documentation of this variable in the Consortium on Safe Labor database.

Each intrapartum prediction model estimated the probability of unfavorable primary outcome (LRS) based on baseline predictors and dynamic labor variables, as well as the most recent LRS estimated using data captured up to the prior examination during labor progression.

## Statistical analysis

All data analyses were performed using the R programming environment for statistical computing version 3.5.1 (R Development Core Team, 2018). We reported descriptive statistics of all covariates in the final sample: mean (standard deviation) was used to summarize continuous variables and counts (percentages) for categorical variables.

**Intrapartum prediction models.** Given that the progress of labor is affected by time-varying (or dynamic) confounders, methods appropriate for adjusting such dynamic confounders are needed to predict maternal and neonatal outcomes more accurately. Methods adopted by Zhang et al [9] were limited in capturing this dynamic aspect of the data. In this study, the use of machine-learning methods capable of capturing representative features from changing labor characteristics is proposed.

Existing analytic methods for labor progression have been based on traditional statistical approaches, which, however, tend to make unrealistic assumptions regarding the functional form of the model and distribution of variables. These assumptions are often not applicable in complex clinical situations such as the dynamic labor process. As a result, the models may not fit the data well and may not be generalizable. Machine-learning methods, on the other hand, can estimate complex relationships between clinical measurements with reasonable accuracy, thus producing robust and consistent estimates without making a priori assumptions. These advanced data analytic techniques have been repeatedly shown to produce astonishing results in many applications in computer sciences, bioinformatics, health care, and elsewhere [10–13]. Thus, in this study, we propose applying machine-learning methods to collectively analyze the patterns of changes in usual prenatal and intrapartum variables based on the large DASH database. Specifically, we implemented an incremental extreme gradient boosting (XGBoost) algorithm [14,15], where starting from the baseline model, the dynamic labor variables (at cervical dilation of 4, 5, 6, 7, 8, 9, and 10 cm) are incrementally used to extend the knowledge of an existing XGBoost model.

The (XGBoost) [15] algorithm is a generalized implementation of the gradient boosting machine (GBM) [14] technique with several algorithmic enhancements designed to significantly improve prediction accuracy, training speed and scalability. An important enhancement is the implementation of the of the least absolute shrinkage and selection operator (LASSO) and the ridge (Ridge) regularization methods [16], which are techniques designed to prevent overfitting.

**Handling intrapartum time-varying variables.** Throughout the labor process, pelvic examination variables are repeatedly measured for each patient, and as such they are potentially correlated, which presents a major challenge for most machine-learning models [11]. Therefore, we aggregated the repeated observations for each patient prior to the current dilation to construct each intrapartum prediction model. Specifically, cross-sectional data for each intrapartum prediction model was created by aggregating dynamic variables to 3 variables: the frequency (count), the median, and the last observed value.

We imputed missing values (with ≤30% missing observations) in the data with the random forest imputation method, missForest [17].

## Training and validation

The GBM model requires a number of tuning parameters to be set for optimal performance and to avoid overfitting. Consequently, we set up a grid for each combination of tuning

parameters and the best combination selected in a 10-fold cross validation. In 10-fold cross validation, 1 randomly partitions the data into 10 mutually exclusive subsets (or folds); 9 folds are used for training and the hold-out fold for testing the performance of the model. We repeated the entire procedure 10 times and averaged the performances on all test folds and computed confidence intervals. The workflow of the training and validation procedure is illustrated in the supplementary figure (S1 Fig).

## Results

### Baseline and intrapartum dynamic characteristics

Out of 228,438 delivery episodes that compose the Consortium on Safe Labor database, 66,586 episodes were eligible for this study (S2 Fig). Mean maternal age at admission was 26.95±6.48 years, mean parity was 0.92±1.23, and pre-pregnancy body mass index (BMI) was 25.24±5.58 kg/m$^2$ with a mean of 14.71±5.92 kg weight gain during pregnancy. Race and ethnicity were diverse; 21,155 (31.8%) were white; 23,128 (34.7%) African-American; 14,862 (22.3%) Hispanic; 2,745 (4.1%) Asian/Pacific Islander; 193 (0.3%) multi-racial; 2,072 (3.1%) belonged to other unspecified races; and 2,431 (3.7%) were reported as unknown. Mean gestational age at admission to labor was 39.35±1.13 weeks of gestation. Medical complications of pregnancy included 10,305 (2.0%) diagnosed with pregestational diabetes during that pregnancy; 1,041 (1.6%) diagnosed with gestational diabetes; 1,106 (1.7%) with gestational hypertension; 1,085 (1.6%) with preeclampsia; and 1,085 (1.6%) with chronic hypertension. The rate of prior CD was 2,394 (3.6%) for the entire cohort. Delivery was initiated by labor induction in 31,932 (48.0%) of these episodes. Detailed demographic and clinical characteristics of the study population are shown in Table 1.

### Primary and secondary outcome variables

Unfavorable labor outcomes, based on study definition, were reported in 14,439 (21.68%) of total delivery episodes. Of these, 10,466 (15.7%) deliveries were intrapartum CDs; 2,395 (3.6%) cases were diagnosed with IAI; 1,261 (2.0%) had postpartum hemorrhage; and 3,743 (5.6%) of delivered neonates were admitted to NICU. The incidence of neonatal sepsis and neonatal death were 880 (1.3%) and 49 (0.1%), respectively (S1 Table).

### Predicting labor outcomes

On admission, machine-learning–based prediction models that performed at a sensitivity of 0.69 (95% confidence interval [CI], 0.68–0.70) and a specificity of 0.68 (95% CI, 0.67–0.69) were used to predict unfavorable labor outcome; Area under curve was 0.75 (95% CI, 0.75–0.75) (Table 2). The highest contributing independent variable to this model was parity. Other significant variables included prior CD, maternal age, maternal pre-pregnancy body mass index, height, gestational age at admission, absence of uterine contractions at admission, and maternal weight gain during pregnancy (Fig 1). The diagnostic performance of intrapartum prediction models trended up with advancement of cervical dilation; model sensitivity increased gradually from 0.70 (95% CI, 0.69–0.70) at 4 cm to 0.79 (95% CI, 0.78–0.80) at 10 cm. Similarly, model specificity rose from 0.72 (95% CI, 0.71–0.73) at 4 cm to 0.84 (95% CI, 0.83–0.85) at 10 cm (Table 2). As shown in Fig 2, Area under curve of intrapartum prediction models at 4, 6, 8, and 10 cm reflected a similar trend (0.78 [95% CI, 0.77–0.78] at 4 cm; 0.89 [95% CI, 0.89–0.90] at 10 cm). The most substantial variable for all intrapartum models was prior risk score from the previous model. Other contributing factors to these models included cervical dilation at last examination, number of cervical examinations, current head station,

**Table 1. Characteristics of eligible patients.**

| Variables[a] | Patients With Favorable Outcomes (n = 52,147) | Patients With Unfavorable Outcomes (n = 14,439) | All Patients (N = 66,586) | P Value |
|---|---|---|---|---|
| Maternal age, years | 26.80±6.40 | 27.47±6.73 | 26.95±6.48 | < .001 |
| Parity | 1.03±1.26 | 0.52±1.01 | 0.92±1.23 | < .001 |
| History of macrosomia in previous pregnancies | 850 (1.6) | 115 (0.8) | 965 (1.4) | < .001 |
| Prepregnancy BMI, kg/m² | 25.05±5.42 | 25.94±6.05 | 25.24±5.58 | < .001 |
| Pregestational diabetes | 941 (1.8) | 364 (2.5) | 1,305 (2.0) | < .001 |
| History of heart disease | 473 (0.9) | 127 (0.9) | 600 (0.9) | .757 |
| Antenatal-positive GBS status | 10,852 (20.8) | 3,103 (21.5) | 13,955 (21.0) | .076 |
| Smoking | 2,674 (5.1) | 651 (4.5) | 3,325 (5.0) | .003 |
| Cerclage placement in current pregnancy | 111 (0.2) | 28 (0.2) | 139 (0.2) | .659 |
| Gestational hypertension | 796 (1.5) | 310 (2.1) | 1,106 (1.7) | < .001 |
| Preeclampsia | 711 (1.4) | 374 (2.6) | 1,085 (1.6) | < .001 |
| Eclampsia | 31 (0.1) | 9 (0.1) | 40 (0.1) | .900 |
| Superimposed preeclampsia | 364 (0.7) | 212 (1.5) | 576 (0.9) | < .001 |
| Chronic hypertension | 549 (1.1) | 229 (1.6) | 778 (1.2) | < .001 |
| Gestational diabetes | 725 (1.4) | 316 (2.2) | 1041 (1.6) | < .001 |
| Intrauterine growth restriction | 292 (0.6) | 79 (0.5) | 371 (0.6) | .855 |
| Oligohydramnios | 967 (1.9) | 413 (2.9) | 1,380 (2.1) | < .001 |
| Polyhydramnios | 74 (0.1) | 43 (0.3) | 117 (0.2) | < .001 |
| Maternal weight on admission, kg | 81.43±16.29 | 84.00±17.79 | 81.99±16.66 | < .001 |
| Gestational age on admission | 39.31±1.11 | 39.50±1.17 | 39.35±1.13 | < .001 |
| Maternal ethnicity | | | | < .001 |
| White | 16,807 (32.2) | 4,348 (30.1) | 21,155 (31.8) | |
| Black | 18,055 (34.6) | 5,073 (35.1) | 23,128 (34.7) | |
| Hispanic | 11,707 (22.4) | 3,155 (21.9) | 14,862 (22.3) | |
| Asian/Pacific Islander | 2,054 (3.9) | 691 (4.8) | 2,745 (4.1) | |
| Multi-racial | 153 (0.3) | 40 (0.3) | 193 (0.3) | |
| Other | 1,567 (3.0) | 505 (3.5) | 2,072 (3.1) | |
| Unknown | 1,804 (3.5) | 627 (4.3) | 2,431 (3.7) | |
| Maternal height, m | 1.63±0.07 | 1.62±0.07 | 1.63±0.07 | < .001 |
| Alcohol use | 1,134 (2.2) | 291 (2.0) | 1,425 (2.1) | .242 |
| Weight change during pregnancy, kg | 14.47±5.82 | 15.58±6.20 | 14.71±5.92 | < .001 |
| ECV in this pregnancy | 92 (0.2) | 16 (0.1) | 108 (0.2) | .083 |
| Pre-pregnancy weight, kg | 66.95±15.68 | 68.39±17.27 | 67.26±16.05 | < .001 |
| Fetal sex | | | | < .001 |
| Female | 26,164 (50.2) | 6,568 (45.5) | 32,732 (49.2) | |
| Male | 25,932 (49.7) | 7,836 (54.3) | 33,768 (50.7) | |
| Ambiguous | 1 (0.0) | 1 (0.0) | 2 (0.0) | |
| Unknown | 50 (0.1) | 34 (0.2) | 84 (0.1) | |
| Previous CDs | | | | < .001 |
| 0 | 50,683 (97.2) | 13,509 (93.6) | 64,192 (96.4) | |
| 1 | 1,420 (2.7) | 833 (5.8) | 2,253 (3.4) | |
| 2 | 39 (0.1) | 87 (0.6) | 126 (0.2) | |
| Induction of labor | 23,586 (45.2) | 8,346 (57.8) | 31,932 (48.0) | < .001 |
| Meconium stained amniotic fluid | | | | < .001 |
| No | 47,375 (90.8) | 12,422 (86.0) | 59,797 (89.8) | |
| Yes (unspecified) | 4,639 (8.9) | 1,954 (13.5) | 6,593 (9.9) | |

*(Continued)*

**Table 1.** (Continued)

| Variables[a] | Patients With Favorable Outcomes (n = 52,147) | Patients With Unfavorable Outcomes (n = 14,439) | All Patients (N = 66,586) | P Value |
|---|---|---|---|---|
| Thin | 81 (0.2) | 34 (0.2) | 115 (0.2) | |
| Moderate | 1 (0.0) | 2 (0.0) | 3 (0.0) | |
| Thick | 51 (0.1) | 27 (0.2) | 78 (0.1) | |
| Method of labor induction | | | | |
| AROM | 1,292 (2.5) | 268 (1.9) | 1,560 (2.3) | < .001 |
| Prostaglandin E1 | 1,067 (2.0) | 719 (5.0) | 1,786 (2.7) | < .001 |
| Mechanical methods | 43 (0.1) | 41 (0.3) | 84 (0.1) | < .001 |
| Prostaglandin E2 | 412 (0.8) | 148 (1.0) | 560 (0.8) | .006 |
| Oxytocin | 12,427 (23.8) | 3,952 (27.4) | 16,379 (24.6) | < .001 |
| Method of ROM | | | | < .001 |
| AROM | 30,380 (58.3) | 8,275 (57.3) | 38,655 (58.1) | |
| SROM | 20,012 (38.4) | 5,713 (39.6) | 25,725 (38.6) | |
| PROM | 14 (0.0) | 8 (0.1) | 22 (0.0) | |
| Others | 356 (0.7) | 46 (0.3) | 402 (0.6) | |
| Unknown | 1,385 (2.7) | 397 (2.7) | 1,782 (2.7) | |

Abbreviations: AROM, artificial rupture of membranes; BMI, body mass index; CD, cesarean delivery; ECV, external cephalic version; GBS, group B streptococci; PROM, prelabor rupture of membranes; ROM, rupture of membranes; SROM, spontaneous rupture of membranes.

[a] Continuous variables are presented as means ± standard deviation; categorical variables are presented as number and percentages.

cervical dilation change, current cervical dilation, and dilation delta. The spectrum of contributing factors and the magnitude of their contribution to baseline and intrapartum prediction models are shown in Fig 1.

LRS was plotted against cervical dilation to demonstrate the LRS trend among women who had favorable versus unfavorable composite outcome (S3 Fig). Women with unfavorable composite outcome had a baseline LRS score above 35%. Their scores at 4 to 6 cm were between 45% and 50% and consistently trended up beyond 60% over increasing cervical dilation. Baseline LRS scores were below 25% among women with favorable composite outcome. The scores trended down from 23% at 4 cm, to 20% at 7 cm, to 15% at 10 cm. Similarly, risk of failed vaginal delivery trended up from 34% on admission to 72% at 10 cm in women delivering by intrapartum CD. In women who had successful vaginal delivery, the risk of failed vaginal delivery was below 20% and trended below 10% at 10 cm (S3 Fig).

## Discussion

The Consortium on Safe Labor database is a multicenter observational database that is comprised of 228,438 deliveries. Utilizing this database, this study applied machine-learning algorithms to generate a series of prediction models that incorporates both static and dynamic predictors, including patient baseline characteristics, most recent clinical assessment, and cumulative labor progress from admission. These models may provide an alternative to current practice, which endorses the use of labor charts. In contrast to labor charts, which set constant margins to safe labor course, machine-learning models promote individualization of clinical decisions using baseline and labor characteristics of each patient.

For several decades, Friedman's sloping curve was cited as a reference of normal labor progress [18]. The terms "latent labor" and "active labor" were introduced in the literature to discriminate initial slow interval (<3–3.5 cm) from subsequent accelerated labor course. In 1972,

**Table 2. Diagnostic performance of machine-learning–based prediction models of unfavorable labor outcomes and intrapartum cesarean delivery at first stage of labor [a].**

| Outcome | Cervical Dilation (in cm) | Error | AUC | Sensitivity | Specificity | PPV |
|---|---|---|---|---|---|---|
| Composite outcome (unfavorable labor outcomes) | Baseline | 0.31 (0.31–0.32) | 0.75 (0.75–0.75) | 0.69 (0.68–0.70) | 0.68 (0.67–0.69) | 0.42 (0.42–0.42) |
| | 4 | 0.29 (0.29–0.30) | 0.78 (0.77–0.78) | 0.70 (0.69–0.70) | 0.72 (0.71–0.73) | 0.50 (0.49–0.51) |
| | 5 | 0.28 (0.28–0.28) | 0.80 (0.80–0.80) | 0.70 (0.70–0.71) | 0.74 (0.73–0.75) | 0.52 (0.52–0.53) |
| | 6 | 0.27 (0.26–0.27) | 0.81 (0.81–0.81) | 0.72 (0.70–0.73) | 0.75 (0.74–0.77) | 0.55 (0.54–0.55) |
| | 7 | 0.25 (0.25–0.26) | 0.83 (0.82–0.83) | 0.73 (0.72–0.74) | 0.76 (0.75–0.77) | 0.56 (0.55–0.57) |
| | 8 | 0.25 (0.24–0.25) | 0.84 (0.83–0.84) | 0.75 (0.74–0.76) | 0.75 (0.74–0.77) | 0.56 (0.54–0.57) |
| | 9 | 0.24 (0.24–0.24) | 0.85 (0.84–0.85) | 0.76 (0.75–0.77) | 0.76 (0.76–0.77) | 0.57 (0.56–0.57) |
| | 10 | 0.19 (0.18–0.19) | 0.89 (0.89–0.90) | 0.79 (0.78–0.80) | 0.84 (0.83–0.85) | 0.67 (0.66–0.68) |
| Intrapartum cesarean delivery | Baseline | 0.29 (0.29–0.30) | 0.78 (0.77–0.78) | 0.71 (0.70–0.72) | 0.70 (0.69–0.71) | 0.37 (0.36–0.37) |
| | 4 | 0.27 (0.26–0.27) | 0.81 (0.81–0.82) | 0.72 (0.71–0.74) | 0.74 (0.73–0.75) | 0.46 (0.45–0.47) |
| | 5 | 0.24 (0.24–0.24) | 0.84 (0.84–0.84) | 0.75 (0.75–0.76) | 0.76 (0.76–0.77) | 0.49 (0.48–0.49) |
| | 6 | 0.23 (0.22–0.23) | 0.86 (0.85–0.86) | 0.76 (0.74–0.77) | 0.79 (0.78–0.79) | 0.52 (0.51–0.53) |
| | 7 | 0.21 (0.21–0.22) | 0.87 (0.87–0.88) | 0.78 (0.78–0.79) | 0.79 (0.78–0.80) | 0.53 (0.52–0.54) |
| | 8 | 0.20 (0.20–0.20) | 0.88 (0.88–0.89) | 0.78 (0.77–0.79) | 0.82 (0.81–0.83) | 0.56 (0.55–0.57) |
| | 9 | 0.19 (0.18–0.19) | 0.90 (0.90–0.90) | 0.80 (0.79–0.80) | 0.83 (0.82–0.83) | 0.58 (0.57–0.59) |
| | 10 | 0.12 (0.11–0.12) | 0.95 (0.95–0.95) | 0.87 (0.86–0.88) | 0.90 (0.89–0.91) | 0.72 (0.71–0.74) |

Abbreviations: AUC, area under curve; PPV positive predictive value.

[a] Values between brackets present 95% confidence interval.

Philpott and Castle [19,20] proposed the use of "alert lines" and "action lines" to facilitate management of labor through a prospective study of 624 Rhodesian African primigravida women and provided simplified directions to midwives in isolated areas. The WHO partogram was derived from these studies and has served as an important tool in managing labor, especially in low-resource countries.

Although the WHO partogram has been adopted globally to standardize labor care and prevent prolonged labor, the routine use of the WHO partogram has been questioned; a Cochrane review of 3 clinical trials (1,813 patients) comparing partogram to no partogram use did not reveal differences in CD rates, duration of first stage of labor, or Apgar score less than 7 at 5 minutes [8]. Despite its increasing use, the rate of CD has substantially increased in the past 3 decades, reaching 32% of total deliveries in the United States in 2017 [21]. This rising trend has not been associated with a concomitant decline in maternal or neonatal mortality [22]. Furthermore, the current rate of NICU admission among neonates delivered at term is notable, accounting for 4.6% of neonates delivered electively at 39 weeks of gestation [23]. The

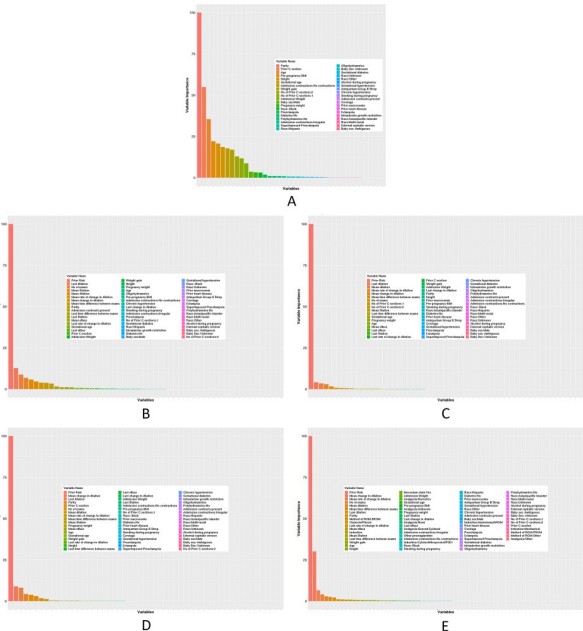

**Fig 1. Baseline and intrapartum predictors of composite unfavorable labor outcome and magnitude of contribution to prediction models.** A, Prediction model on admission. B, Prediction model at 4 cm cervical dilation. C, Prediction model at 6 cm cervical dilation. D, Prediction model at 8 cm cervical dilation. E, Prediction model at 10 cm cervical dilation.

trend is increasing, and term neonates weighing at least 2,500 grams at birth may represent more than 50% of total NICU admissions [24].

Zhang et al [7] hypothesized that current recommendations of management of labor were based on Friedman's study in the 1950s and may not reflect current populations. The new partogram differs from the WHO partogram; the 95th percentile line, which corresponds to the WHO action line, is an exponential-like stair line, which outlines a contemporary course of cervical dilation. Unlike the WHO partogram, which aims to prevent prolonged labor, Zhang et al proposed their partogram as a clinical tool to prevent premature CD without taking into account important maternal and neonatal outcomes. Thus, a secondary analysis of a prospective cohort study of 7,845 women with term low-risk pregnancy from 2010 through 2014 was conducted to assess maternal and neonatal outcomes after implementation of the Zhang labor curves. Rosenbloom et al [25] reported that the primary CD rate did not decline between 2010 and 2014 (15.8% vs 17.7%, $P = 0.5$). In addition, maternal and neonatal morbidity significantly increased in the same time frame. A multicenter, cluster-randomized, controlled trial (LaPS trial) was conducted in 14 clusters in Norway; 7 obstetric units were randomly assigned to intervention (managed by Zhang's guidelines, n = 3,972) versus 7 units that were assigned to control (managed by the WHO partogram, n = 3,305). Again, the rate of intrapartum CD was not significantly different between the 2 groups [26].

In this study, we hypothesized that the challenges associated with creation of labor charts are attributed to more than index population. Labor is a complex physiologic process, and outcomes are likely to be influenced by several factors. These factors can either be identifiable (determined at baseline) or unknown, yet they are indirectly reflected on labor course. Machine-learning algorithms have been increasingly used in data mining scenarios of large databases when the domain is poorly understood or when dynamic models are needed. Compared to conventional statistical methods, machine learning minimizes statistical assumption

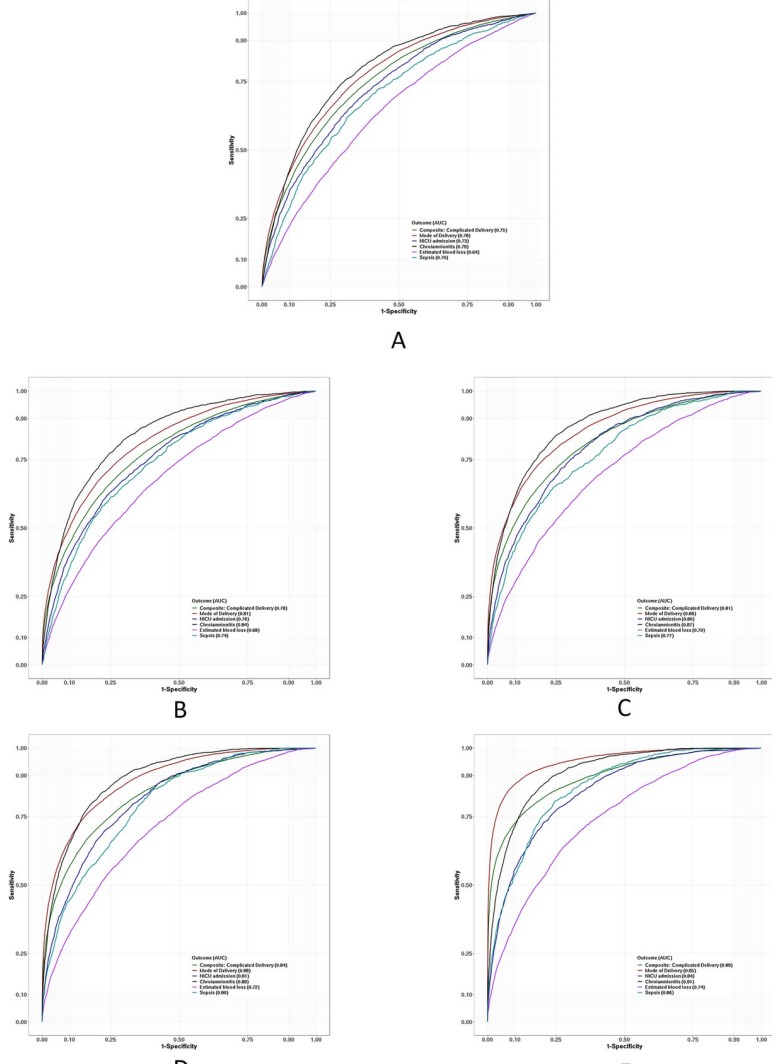

**Fig 2. Diagnostic performance of baseline and intrapartum prediction models for composite unfavorable labor outcome.** A, Area under curve (AUC) on admission. B, AUC at 4 cm cervical dilation. C, AUC at 6 cm cervical dilation. D, AUC at 8 cm cervical dilation. E, AUC at 10 cm cervical dilation. NICU indicates neonatal intensive care unit.

and works by identifying hidden patterns within data and incorporating evolving risks during the labor progression into outcome predictions. Therefore, the predictive power of these models is generally strong [27,28]. In this study, we used a large national database to create a series of dynamic prediction models, as an alternative to conventional labor charts, to predict labor outcomes. These models promote individualized assessment of labor progress based on patient characteristics and current labor patterns. They do not incorporate fixed definitions of latent labor, active labor, or rate of cervical dilation. Alternatively, an LRS graph can be used to determine the cumulative likelihood of safe labor, taking into account the likelihood of CD and any adverse maternal and neonatal outcomes. A patient's baseline LRS, LRS trend over time, and LRS graph in relation to reference LRS graph can improve the intrapartum decision-making process.

To our knowledge, this is the first study that implements machine-learning algorithms in labor management. The study is highly generalizable because it used a large national database

with an ethnically diverse cohort of women that is not restricted by parity, previous CD, or certain maternal or neonatal outcomes. The study is limited by the retrospective nature of collected data. Furthermore, decision of CD obscures outcomes of further expectant management. However, these limitations are inherent in all labor and delivery studies due to ethical concerns of maternal and fetal exposure to unjustifiable risks. Fetal heart rate monitoring was not included in this study due to lack of documentation. Therefore, fetal heart tracing should be interpreted independently, and response to abnormal findings should be made per protocol. Other potential limitation of our study is the definition of the composite outcome, which is comprised of heterogenous adverse labor outcomes. However, the clinical rational for constructing this composite outcome is that the occurrence of any of the events in the composite outcome would trigger ending labor and expediting delivery. In addition, summarizing the risk in a single parameter like the one defined through our composite outcome can be easily interpreted by the obstetrician and the patient for counseling and decision-making purposes. Finally, the results of this study cannot be converted to a printed labor chart due to the complexity of machine-learning algorithms. However, a digital application is currently under development to facilitate clinical use of this developing tool.

## Conclusions

In conclusion, utilization of machine-learning–based algorithms may provide a dynamic, cumulative, and individualized model for prediction of outcomes of vaginal delivery and facilitation of intrapartum decision making. LRS charts may be used as an efficient alternative to conventional labor charts. However, further prospective studies are warranted to assess outcomes of implementation of these models in labor units.

## Supporting information

**S1 Fig. Workflow for training and validation of the incremental machine learning model.** Each model (except baseline model) uses labor risk score (LRS) predictions from the previous model. Data were randomly divided into 10 equal and independent parts: The model was trained on 9 folds and validated on the last fold. The procedure was repeated until each fold was used once for validation. At each step, optimal tuning parameters of the model were selected, and performance was evaluated on the validation fold. Overall iterative process was repeated 10 times and performance results were averaged.
(GIF)

**S2 Fig. Flow chart of study cohort.**
(PNG)

**S3 Fig. Trend of Labor Risk Score Over Labor Progress Among Women with Favourable and Unfavourable Labor Outcome.** A, Women with unfavourable (red line) versus favourable (green line) composite labor outcome. B, Women who had cesarean delivery (red line) versus vaginal delivery (green line).
(PNG)

**S1 Table. Unfavourable labor outcomes among study population.**
(DOCX)

## Acknowledgments

The study was conducted using the Eunice Kennedy Shriver National Institute of Child Health and Human Development (NICHD) *Consortium on Safe Labor* database.

## Author Contributions

**Conceptualization:** Sherif A. Shazly, Abimbola O. Famuyide.

**Data curation:** Sherif A. Shazly, Abimbola O. Famuyide.

**Formal analysis:** Bijan J. Borah, Che G. Ngufor.

**Supervision:** Abimbola O. Famuyide.

**Writing – original draft:** Sherif A. Shazly, Bijan J. Borah, Che G. Ngufor, Abimbola O. Famuyide.

**Writing – review & editing:** Vanessa E. Torbenson, Regan N. Theiler.

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
