## [Decision Letter · Decision Letter 0]

27 Apr 2022

PONE-D-22-07903Impact of Labour Characteristics on Maternal and Neonatal Outcomes of Labour:  A Machine-Learning ModelPLOS ONE

Dear Dr. Famuyide,

Thank you for submitting your manuscript to PLOS ONE. After careful consideration, we feel that it has merit but does not fully meet PLOS ONE’s publication criteria as it currently stands. Therefore, we invite you to submit a revised version of the manuscript that addresses the points raised during the review process.

We look forward to receiving your revised manuscript.

Kind regards,

Jonas Bianchi, DDD, MS, Ph.D

Academic Editor

PLOS ONE

Journal Requirements:

Reviewers' comments:

Reviewer's Responses to Questions

**Comments to the Author**

1. Is the manuscript technically sound, and do the data support the conclusions?

Reviewer #2: Partly

Reviewer #3: Yes

Reviewer #4: Yes

2. Has the statistical analysis been performed appropriately and rigorously? 

Reviewer #2: Yes

Reviewer #3: Yes

Reviewer #4: Yes

3. Have the authors made all data underlying the findings in their manuscript fully available?

Reviewer #2: No

Reviewer #3: Yes

Reviewer #4: Yes

4. Is the manuscript presented in an intelligible fashion and written in standard English?

Reviewer #2: Yes

Reviewer #3: Yes

Reviewer #4: Yes

5. Review Comments to the Author

Reviewer #2: The authors used machine learning models to predict adverse pregnancy outcomes during labor. Although their study is well presented and the statistical methods rigorous and apparently adequate i believe that the incorporation of several adverse outcomes that mix up maternal and fetal/neonatal events does not help physicians to interprete the provided results. Furthermore, the variable that were used as predictive are unclear as the authors do not present them in their study. I believe that the authors should evaluate separately the adverse pregnancy outcomes that are related to maternal and fetal/neonatal adverse events and incorporate variables that are pathophysiologically correlated to these outcomes. Given the sample size of their population this seems to be powered enough and may provide a trully intelligent and usefull information for physicians.

Reviewer #3: Abstract:

Abstract is succinct, but I recommend adding that this is a gradient boosting machine learning model to be more specific.

Conclusion is very general, I suggest making it more relevant to the presented data.

Introduction:

Line 48- I would recommend rewording the sentence about Friedman as it was not a 'trial,' which implies a randomized controlled trial.

Materials and Methods:

Why did the authors define their composite this way? It seems very heterogeneous at it includes labor outcomes, maternal morbidities and neonatal morbidities. If the authors could imagine counseling a patient on these risks, would this type of composite information make it easier or more difficult to counsel someone? Could the authors predict these outcomes individually? Or group them by mother and baby? This may be more clinically relevant.

How was the composite calculated? Is this a linear calculation? Were variables weighted?

Why is IAI and outcome of interest? I would include this as a baseline characteristic because it develops during labor.

Why chose meconium as a baseline variable? This has not been used in the US clinically to make decisions for a few decades. Did it make it into the model? Is this to appeal to an international audience?

The authors should still have submitted this work to their IRB for exempt determination status.

Why include multiparous women? Most adverse labor outcomes happen in nulliparas and this might make your model more specific.

Did you include patients with and without epidural anesthesia?

Results:

Interesting analysis of accuracy with each advancement of cervical dilation. How did the authors account for time? How did they account for women with a repeat cervical exam of the same dilation as the one prior?

How many exams were recorded per patient, on average?

Need more discussion of the LRS score in the Methods section. What does this mean exactly? Does a labor risk score = risk of adverse labor outcome? This is a bit unclear.

Discussion:

How does one use this tool? Is it online? on paper? I see that the authors discuss a digital application development. Given that the authors discuss the WHO partogram in detail, will the authors also develop something that would be universally accusable? Ie, if OBGYN providers dont have smartphones or wifi to access an app, how can this model be useful?

Line 287- please cite: Gimovsky AC, Levine JT, Pham A, Dunn J, Zhou D, Peaceman AM. Pushing the bounds of second stage in term nulliparas with a predictive model. Am J Obstet Gynecol MFM. 2019 Aug;1(3):100028. doi: 10.1016/j.ajogmf.2019.07.001. Epub 2019 Jul 20. PMID: 33345792.

I recommend adding more info about gradient boosting- ie, why choose this type of model, how does it compare to other machine learning models, etc.

Tables:

Are the headings correct in Table 1? ie more patients had unfavorable (52,147) outcomes than favorable (14,439)? I think these columns might be mislabeled. As it reads now you have a better outcome is you are older, less parous (would change to median/standard error, as parity is an integer), have diabetes, hypertension, preeclampsia, oligohydramnios, etc....

Minor: As Plos One is an American journal I suggest the use of American English spellings... ie "labor", not "labour"; foetal = fetal, etc.

The authors should review the manuscript for several typos and grammatical errors.

Reviewer #4: In this study Authors evaluated the performance of a ML model in predicting labor outcome from data retrospectively analyzed form a large database. The subject is of interest and I would like to congratulate with Authors for their effort

My comments are

1)the definition of unfavorable outcome was really heterogeneous, In other words Authors included variables with different pathogenesis such as emergency CS and postpartum hemorrhage for which the constructed model may have a different performance. Since the database is relatively large I strongly suggest too construct individual model for each outcome variable

2) the evaluation of the model according to cervical dilation is of interest but clinically speaking is only one of the variable that influence labor outcome. Have Authors concomitant data on fetal head station, occiput position and duration on labor? I guess that some of these data are present in. the database and should be used

3)parity is a crucial point in predicting labor outcome. So different models should be used for nulli and para women

6. PLOS authors have the option to publish the peer review history of their article (what does this mean?). If published, this will include your full peer review and any attached files.

Reviewer #1: No

Reviewer #2: No

Reviewer #3: No

Reviewer #4: **Yes: **Giuseppe Rizzo

---

## [Author Response · Author response to Decision Letter 0]

29 Jun 2022

Review Comments to the Author

Reviewer #2: The authors used machine learning models to predict adverse pregnancy outcomes during labor. Although their study is well presented and the statistical methods rigorous and apparently adequate i believe that the incorporation of several adverse outcomes that mix up maternal and fetal/neonatal events does not help physicians to interprete the provided results. Furthermore, the variable that were used as predictive are unclear as the authors do not present them in their study. I believe that the authors should evaluate separately the adverse pregnancy outcomes that are related to maternal and fetal/neonatal adverse events and incorporate variables that are pathophysiologically correlated to these outcomes. Given the sample size of their population this seems to be powered enough and may provide a trully intelligent and usefull information for physicians.

Response: Thank you for your valuable comments:

● We appreciate the reviewer’s comment about having multiple heterogenous adverse labor outcomes included in our composite outcome. However, the clinical rationale for constructing this composite outcome is that the occurrence of this composite outcome would recommend ending labor. Summarizing the risk in a single parameter like the one defined through our composite outcome can be easily interpreted by the obstetrician and the patient for counseling and decision-making purposes. Please note that a follow-up study is currently in progress to use this model to predict probability of individual outcomes. Nevertheless, we acknowledge this is a potential limitation in the Discussion Section of the manuscript. It is important, though, to emphasize that this information is descriptive of the model and should not be used to indicate a direct correlation between a variable and an outcome which is not the intent of machine learning models.

● Regarding variables used, all variables were plotted in Figure 1 according to their magnitude of impact on model outcome

● Since these models were created using machine learning algorithms, selection of variables using our knowledge of a pathophysiological correlation, would impact the performance of machine learning algorithms. Technically, we do not intervene in variable selection to allow machine learning to recognize the hidden interactions between variables, some of which may not be known to affect outcome. This is one of the key features of ML vs. conventional statistics in establishing prediction models, which does not apply if the indication of the study is to investigate an association between a variable(s) and an outcome.

Reviewer #3: Abstract:

Abstract is succinct, but I recommend adding that this is a gradient boosting machine learning model to be more specific.

Response: Thank you. We added model specification to the abstract (abstract: under materials and methods)

Conclusion is very general, I suggest making it more relevant to the presented data.

Response: we changed “the conclusion” to be more specific of the results of the study (abstract: under conclusion) 

Introduction:

Line 48- I would recommend rewording the sentence about Friedman as it was not a 'trial,' which implies a randomized controlled trial.

Response: The word “trials” was corrected to “studies” (highlighted in yellow)

Materials and Methods:

Why did the authors define their composite this way? It seems very heterogeneous at it includes labor outcomes, maternal morbidities and neonatal morbidities. If the authors could imagine counseling a patient on these risks, would this type of composite information make it easier or more difficult to counsel someone? Could the authors predict these outcomes individually? Or group them by mother and baby? This may be more clinically relevant.

Response: We used a composite outcome of adverse labor events, occurrence of any of which would recommend against continuation of labor, to provide a single parameter that can be easily interpreted by the obstetrician and the patient for counseling and decision-making purposes. Each component of our composite outcome by itself can potentially inform against a decision to proceed with normal delivery. Thus, obstetricians and patients would be informed and be able to relate to the fact that there was a strong likelihood of a major adverse event if labor continues beyond a particular point. In summary, setting a single parameter was meant to be the trigger for potential intervention/counseling. However, an application is currently in progress to use this model to predict probability of individual outcomes.

How was the composite calculated? Is this a linear calculation? Were variables weighted?

Response: We did not weight the components in the definition of our composite outcome. The composite outcome was deemed to occur if any of the unfavorable labor outcomes described in the “Study Outcomes” of the manuscript occurred. As explained above, the rationale for forming the composite in this way was that the occurrence of any of those unfavorable outcomes would recommend against continuation of labor, and therefore we did not think providing differential weights for the components was relevant. 

Why is IAI and outcome of interest? I would include this as a baseline characteristic because it develops during labor.

Response: We consider IAI an outcome of interest rather than a baseline variable since women were included early in labor where no IAI had developed yet. IAI is a considerable event that usually develops late in labor and could likely present a complication of prolonged labor course. So, it was important for us to consider it as an outcome to balance against prolonged expectancy for the sake of achieving vaginal delivery as a sole target.

Why chose meconium as a baseline variable? This has not been used in the US clinically to make decisions for a few decades. Did it make it into the model? Is this to appeal to an international audience?

Response: since we used machine learning algorithms to develop these models, we did not select variables to be included in the model. Machine learning algorithms ideally work by feeding them with all available variables, even if not apparently clinically relevant, since they can recognize hidden interactions among variables which would collectively impact outcome. Including a variable in the algorithm does not mean that this variable would impact the outcome and the algorithm will decide whether it would or would add to model predictability and to what extent. This eliminates a source of bias that besets conventional statistical analysis. It is important to highlight that machine learning models are used to predict an outcome based on a set of variables and how they interact and should not be used to establish an association or causality between a single variable and an outcome, which is better investigated using conventional statistics.

As for “meconium” specifically, it does not merely guide a decision and is considered in the context of other factors to decide if it, at all, would change the probability of an outcome. This hypothesis appears to be true in our data as the impact of meconium in our model appears to be only moderate as is shown in Figure 1-E. 

The authors should still have submitted this work to their IRB for exempt determination status.

Thanks for this comment. We submitted our work to Mayo Clinic IRB for review, and because our dataset is completely deidentified, Mayo IRB determined that our study does not require a review. We have included a sentence at the beginning of the Methods section to this effect. Please also see the attached memo from Mayo IRB.

Why include multiparous women? Most adverse labor outcomes happen in nulliparas and this might make your model more specific.

Response: We included multipara to ensure generalizability of results. Multipara still would benefit from a chart or a score, which would ensure their labor progress and management is within the safe limit. In fact, current labor charts are used for both primigravida and multipara although they were created using data from primigravida, and there are no labor charts that are designed for higher parity. 

Understanding that nullipara is specifically at higher risk of complications, the model also recognizes that, and treats parity as one of the variables before calculating the probability of an outcome. Therefore, including multipara to the model adds to its generalizability and should not impact its capacity to calculate probability of adverse outcomes in nullipara.

Did you include patients with and without epidural anesthesia?

Response: Yes. Type of anesthesia was not used as a selection criterion.

Results:

Interesting analysis of accuracy with each advancement of cervical dilation. How did the authors account for time? How did they account for women with a repeat cervical exam of the same dilation as the one prior?

Response: Time was documented in the database and was calculated using time of birth as a 00:00 point. Thus, each cervical dilation/examination was documented against time of birth and time between exams was simple to calculate.

All repeat exams were considered as new exams, and thus a new observation in our data. In situations where the dilation remained constant from prior exam(s), however, since other patient factors may have changed since last exam, the repeated exams automatically become new observations (new rows) in our data. 

By considering an incremental XGBoost model, that takes into account prior learned knowledge by the model, this may help to correct for any correlation that may exists between the repeated observations for the same subject. 

How many exams were recorded per patient, on average?

Our data indicate an average of 4 exams. A few women had as many as 14 exams! 

Need more discussion of the LRS score in the Methods section. What does this mean exactly? Does a labor risk score = risk of adverse labor outcome? This is a bit unclear.

Response: That is correct. So, LRS is just a term for ease of use, which indicates the score or the probability of labor adverse outcome, as calculated by the model. We modified the methodology section to clarify this point (highlighted in yellow: under study outcomes).

Discussion:

How does one use this tool? Is it online? on paper? I see that the authors discuss a digital application development. Given that the authors discuss the WHO partogram in detail, will the authors also develop something that would be universally accusable? Ie, if OBGYN providers dont have smartphones or wifi to access an app, how can this model be useful?

Response: Machine learning models typically use complex calculations to predict probability of an outcome implying that it is not feasible to use them using simple equations. Therefore, a paper form is not ideal to use them. Thus, these models are used through an application that treats simple inputs provided by the user to calculate the score. These applications can be available through smart phone, and online. An alternative could be an offline application which would work on any PC, even if not connected to the internet. We are working to develop these additional components in the next few months. 

Line 287- please cite: Gimovsky AC, Levine JT, Pham A, Dunn J, Zhou D, Peaceman AM. Pushing the bounds of second stage in term nulliparas with a predictive model. Am J Obstet Gynecol MFM. 2019 Aug;1(3):100028. doi: 10.1016/j.ajogmf.2019.07.001. Epub 2019 Jul 20. PMID: 33345792.

Response: Reference added (highlighted in yellow [27])

I recommend adding more info about gradient boosting- ie, why choose this type of model, how does it compare to other machine learning models, etc.

Response: Thank you. We have added a brief description of the extreme gradient boosting algorithm used to develop our LRS incremental model. 

Tables:

Are the headings correct in Table 1? ie more patients had unfavorable (52,147) outcomes than favorable (14,439)? I think these columns might be mislabeled. As it reads now you have a better outcome is you are older, less parous (would change to median/standard error, as parity is an integer), have diabetes, hypertension, preeclampsia, oligohydramnios, etc....

Response: Thanks for catching this, the columns were mislabeled and have been corrected. 

Minor: As Plos One is an American journal I suggest the use of American English spellings... ie "labor", not "labour"; foetal = fetal, etc.

The authors should review the manuscript for several typos and grammatical errors.

Response: we have reviewed and modified the manuscript using American English style

Reviewer #4: In this study Authors evaluated the performance of a ML model in predicting labor outcome from data retrospectively analyzed form a large database. The subject is of interest and I would like to congratulate with Authors for their effort

My comments are

1)the definition of unfavorable outcome was really heterogeneous, In other words Authors included variables with different pathogenesis such as emergency CS and postpartum hemorrhage for which the constructed model may have a different performance. Since the database is relatively large I strongly suggest too construct individual model for each outcome variable

Response: Please see the responses to the reviewer # 2 above on this point. In essence, we used a composite outcome of adverse labor events, which would recommend against continuation of labor, to provide a single parameter that can be easily interpreted by the obstetrician and the patient for counseling and decision-making purposes. However, a study is currently in progress to use this model to predict probability of individual outcomes. Thus, obstetricians and patients would be able to recognize what would be the major concern if labor continues beyond a particular point. So, in summary, setting a single parameter was meant to be the trigger for potential intervention/counseling

2) the evaluation of the model according to cervical dilation is of interest but clinically speaking is only one of the variable that influence labor outcome. Have Authors concomitant data on fetal head station, occiput position and duration on labor? I guess that some of these data are present in. the database and should be used

Response: Yes. We included all data, related to the examination, as variables in the models, including fetal head station, cervical effacement and membrane status, are used to predict outcomes. However, we did not use head position since it was not routinely commented on in the database during the first stage. Since occiput position is not routinely checked in the first stage of labor, it would be challenging to document for probability calculation if included in our model. 

3)parity is a crucial point in predicting labor outcome. So different models should be used for nulli and para women

Response: Parity is a major factor in predicting clinical outcomes in relation to course. We included parity as one of the variables, which means that it is considered, in association with other interacting clinical factors, in model prediction, and it contributes significantly to prediction as shown in figure 1. Including parity as a variable also covers for a wider range of parity without needing to split the database since there would be a chance that labor course would differ between a P1 and higher parities or between low parity and high parity (e.g., P4, 5 or more)

---

## [Decision Letter · Decision Letter 1]

4 Aug 2022

Impact of Labor Characteristics on Maternal and Neonatal Outcomes of Labor:  A Machine-Learning Model

PONE-D-22-07903R1

Dear Dr. Famuyide,

We’re pleased to inform you that your manuscript has been judged scientifically suitable for publication and will be formally accepted for publication once it meets all outstanding technical requirements.

Kind regards,

Jonas Bianchi, DDD, MS, Ph.D

Academic Editor

PLOS ONE

---

## [Editor Report · Acceptance letter]

12 Aug 2022

PONE-D-22-07903R1 

Impact of Labor Characteristics on Maternal and Neonatal Outcomes of Labor: A Machine-Learning Model 

Dear Dr. Famuyide:

I'm pleased to inform you that your manuscript has been deemed suitable for publication in PLOS ONE. Congratulations! Your manuscript is now with our production department. 

Kind regards, 

on behalf of

Dr. Jonas Bianchi 

Academic Editor

PLOS ONE